# Automated MRI system for clinically significant prostate cancer detection development validation and real-world implementation

Hanchang Wu [1,12], Fang Liu [1,12], Qingsong Yang[2,12], Haihu Chen[2,12], Yan Wang[3,12], Xiaoguang Yang[4,12], Peng Xia[5,12], Lei Fang[6,12], Gang Li[7], Jing Yang[8], Yindeng Luo[8], Jing Li[1], Xu Fang[1], Xuedong Yang[9], Hui Jiang[10], Jianying Liu[11], Yusi Yi[11], Xiangfei Chai[11], Jianping Lu [1], Jian Wang[1], Xu Gao [3 ✉], Chengwei Chen [1 ✉], Chengwei Shao [1 ✉] & Yun Bian [1 ✉]

Prostate MRI enables detection of clinically significant prostate cancer (csPCa), yet variability in PI-RADS scoring limits reproducibility and throughput. Here, we report the development and validation of an automated MRI-based decision aid (ProAI) that estimates patient-level risk of csPCa from biparametric MRI and supports routine reporting. Training, internal validation, and external testing spanned 7849 examinations across six centres and two public datasets. On pooled external tests, the system achieved a patient-level AUC of 0.93 (95% CI, 0.91–0.95), comparable to PI-RADS while improving inter-case consistency. In a multi-reader, multi-case study involving nine clinicians, assistance increased accuracy from 0.80 to 0.86 and reduced reading time. Prospective implementation in 1978 consecutive examinations-maintained performance (AUC 0.92) and was associated with a 32% reduction in radiology workload. Performance generalised to the TCIA cohort (AUC 0.83). These findings indicate that an automated MRI-based decision aid can standardise reporting and enhance efficiency across prostate cancer care pathways. This study was registered at ClinicalTrials. Trial number: ChiCTR2400092863.

Prostate cancer represents the second most common cancer among men worldwide, with over 1.4 million new cases diagnosed annually and substantial mortality in advanced stages[1]. The clinical challenge lies in distinguishing clinically significant prostate cancer (csPCa), which requires immediate intervention, from indolent disease that can be safely monitored through active surveillance[1,2]. This distinction is critical for preventing both overtreatment of low-risk disease and delayed intervention for aggressive tumors that may progress to metastatic disease.

[1]Department of Radiology, Changhai Hospital, Naval Medical University, Shanghai, China. [2]Department of Intervention, Changhai Hospital, Naval Medical University, Shanghai, China. [3]Department of Urology, Changhai Hospital, Naval Medical University, Shanghai, China. [4]Department of Radiology, The First Hospital of Hohhot City, Inner Mongolia Autonomous Region, Hohhot, China. [5]Department of Radiology, Wuxi Traditional Chinese Medicine Hospital, Wuxi, China. [6]Department of Radiology, Pudong Hospital, Shanghai, China. [7]Department of Radiology, Shandong Qianfoshan Hospital, Jinan, China. [8]Department of Radiology, The Second Affiliated Hospital of Chongqing Medical University, Chongqing, China. [9]Department of Radiology, Shanghai 411 Hospital, Shanghai University, Shanghai, China. [10]Department of Pathology, Changhai Hospital, Naval Medical University, Shanghai, China. [11]Huiying Medical Technology (Beijing) Co. Ltd, Beijing, China. [12]These authors contributed equally: Hanchang Wu, Fang Liu, Qingsong Yang, Haihu Chen, Yan Wang, Xiaoguang Yang, Peng Xia, Lei Fang. ✉e-mail: gaoxu.changhai@foxmail.com; timchen91@aliyun.com; chengweishaoch@163.com; bianyun2012@foxmail.com

Multiparametric magnetic resonance imaging (mpMRI) with Prostate Imaging and Reporting Data System (PI-RADS) scoring has emerged as the standard diagnostic approach for csPCa detection[3,4]. However, significant limitations persist in clinical practice. PI-RADS interpretation exhibits substantial inter-observer variability, with expert agreement rarely exceeding κ = 0.7, creating diagnostic inconsistency across institutions[5-7]. Performance characteristics remain suboptimal, particularly at intermediate risk categories where clinical decisions are most challenging—PI-RADS 3 lesions show poor specificity (<0.80) while PI-RADS 4 lesions demonstrate inadequate sensitivity (<0.75)[8-10]. These limitations contribute to unnecessary biopsies, missed diagnoses, and inefficient resource utilization in healthcare systems globally.

Artificial intelligence approaches, particularly deep learning models, have demonstrated remarkable success in medical imaging applications, offering objective, reproducible analysis that could address current diagnostic challenges[11,12]. Recent studies have shown promising technical performance for AI-based csPCa detection, with some models achieving diagnostic accuracy comparable to or exceeding human radiologists[13,14]. However, a critical gap exists between technical validation and clinical implementation. Most studies lack the scale, diversity, and prospective validation necessary to demonstrate real-world clinical utility. Specifically, limited evidence exists regarding AI's impact on physician workflow efficiency, diagnostic confidence, clinical decision-making, and healthcare resource allocation—factors essential for successful clinical translation.

Here, we address this translational gap through the comprehensive development and validation of ProAI, a fully automated deep learning system for csPCa detection. Our study encompasses four distinct phases: (1) multicenter development and technical validation across diverse populations, (2) systematic evaluation of AI-assisted physician performance, (3) assessment of clinical integration patterns and physician acceptance, and (4) prospective implementation in real-world clinical practice. Through this comprehensive approach, we demonstrate not only superior diagnostic performance but also tangible benefits in clinical workflow efficiency and healthcare delivery, providing evidence for the practical value of AI integration in prostate cancer screening.

## Results

### Baseline clinical characteristics
We assembled a multicentre dataset of 7849 patients drawn from six hospitals and two public repositories. Hospital 1 contributed 3272 cases (January 2016–December 2022); PI-CAI provided 1500 consecutive patients (December 2011–August 2021); the external validation cohorts (tests 1–5) included 839 cases (June 2017–December 2022); and the more ethnically diverse TCIA dataset comprised 260 cases (January 2006–February 2011). In addition, the prospective real-world implementation comprised 1978 consecutive examinations (July 2023–June 2024). All datasets adhered to prespecified inclusion and exclusion criteria (Fig. 1). Patient demographics are summarised in Table 1 and Supplementary Note 1. The four-stage development and validation protocol is outlined in Fig. 2 and Supplementary Fig. 1. Where histology was available, International Society of Urological Pathology (ISUP) grades[15] were most commonly 2–3, with fewer 4–5 cases; TCIA contained a higher proportion of lower grades (ISUP 1–2), consistent with its registry-style, cross-ethnicity composition. Follow-up availability varied by cohort: 27.1% in the training set, 0% in the internal validation and TCIA cohorts, approximately 8–27% across external test sites, and 56–58% in the two prospective real-world periods. These differences reflect local workflows and study design but do not affect comparability for the primary endpoints.

### AI architecture innovation and segmentation performance
We evaluated the impact of different segmentation networks—nnUNet, nn-SAM, and LightM-UNet—on model performance to establish optimal prostate and lesion detection capabilities (Fig. 3A–D, Supplementary Table 1). LightM-UNet Using a conservative detection threshold (Dice value > 0.1), ProAI achieved exceptional sensitivity for clinically significant lesions: 96.4% for PI-RADS 4 lesions (133/138) and 98.0% for PI-RADS 5 lesions (200/204). Detection rates for PI-RADS 3 lesions were 87.9% (152/173), demonstrating balanced performance across the clinical spectrum (Supplementary Tables 2, 3). This high detection sensitivity directly supports the model's primary objective of identifying csPCa patients (Fig. 3E). Comparative analysis revealed nnUNet's superiority for downstream classification accuracy (Fig. 3F), with detailed precision analysis provided in Supplementary Note 2.

Importantly, robustness analysis at varying segmentation quality thresholds revealed that ProAI maintained superior diagnostic performance even with suboptimal segmentation (AUC = 0.959 at DSC < 0.3). This finding is clinically significant as 82% of these cases represented pathologically confirmed benign lesions, validating ProAI's discriminative capability independent of perfect segmentation (Supplementary Table 4).

### Superior patient-level diagnostic performance
ProAI demonstrated excellent discrimination across all validation cohorts at the patient level. Training set performance achieved AUC 0.94 (95% CI: 0.93–0.95), with validation set AUC 0.88 (95% CI: 0.85–0.91). External validation demonstrated robust generalizability: test 1 AUC 0.93 (95% CI: 0.90–0.96), test 2 AUC 0.86 (95% CI: 0.80–0.97), test 3 AUC 0.90 (95% CI: 0.79–1.0), test 4 AUC 0.97 (95% CI: 0.84–0.99), and test 5 AUC 0.96 (95% CI: 0.94–0.99). The aggregated external validation performance yielded AUC 0.93 (95% CI: 0.91–0.95). The optimal operating thresholds for ProAI were determined separately for each dataset by maximizing Youden's index, and the PI-RADS cutoff of ≥3 was used for sensitivity/specificity analysis.

Critically, validation using the TCIA dataset demonstrated ProAI's generalizability across diverse populations, achieving AUC 0.83 (95% CI: 0.78–0.88), statistically comparable to PI-RADS performance (AUC 0.85, 95% CI: 0.80–0.89; P = 0.249). While modestly lower than Asian cohort performance (AUC 0.86–0.97), the maintained clinical acceptability (sensitivity 0.75, specificity 0.76) validates cross-ethnicity applicability. Comprehensive performance metrics are detailed in Table 2, Fig. 3G–J, and Supplementary Fig. 2.

### Lesion-level precision and comparative analysis
At the lesion level, ProAI maintained consistent high performance: validation set AUC 0.89 (95% CI: 0.86–0.92), with external tests achieving AUCs ranging from 0.88 to 0.98, culminating in an overall external validation AUC of 0.94 (95% CI: 0.93–0.96) (Supplementary Table 5, Supplementary Fig. 3).

Error analysis of 167 cases (20% of test set) with significant segmentation errors (DSC < 0.4) revealed ProAI's remarkable robustness, maintaining AUC 0.921 despite suboptimal segmentation. Importantly, most segmentation errors occurred in benign patients, demonstrating the model's clinical reliability under real-world conditions (Supplementary Note 3, Supplementary Fig. 4).

Compared with published results on other datasets, ProAI achieved an AUC of 0.93 (95% CI, 0.91–0.95) in our external validation cohorts, which is higher than that reported by Cai et al.[13] (AUC 0.86; 95% CI, 0.80–0.91) and comparable to Saha et al.[14] (AUC 0.91; 95% CI, 0.87–0.94). Because these are indirect, cross-study comparisons on non-identical cohorts, they should be interpreted cautiously. In addition, our study additionally reports multicentre external validation and a pre-specified prospective clinical implementation evaluation.

### Technical robustness across imaging protocols
Subgroup analyses revealed imaging quality affects both AI and PI-RADS performance; in certain lower-resolution subgroups, PI-RADS shows smaller AUC drops, whereas the AI model is more sensitive to

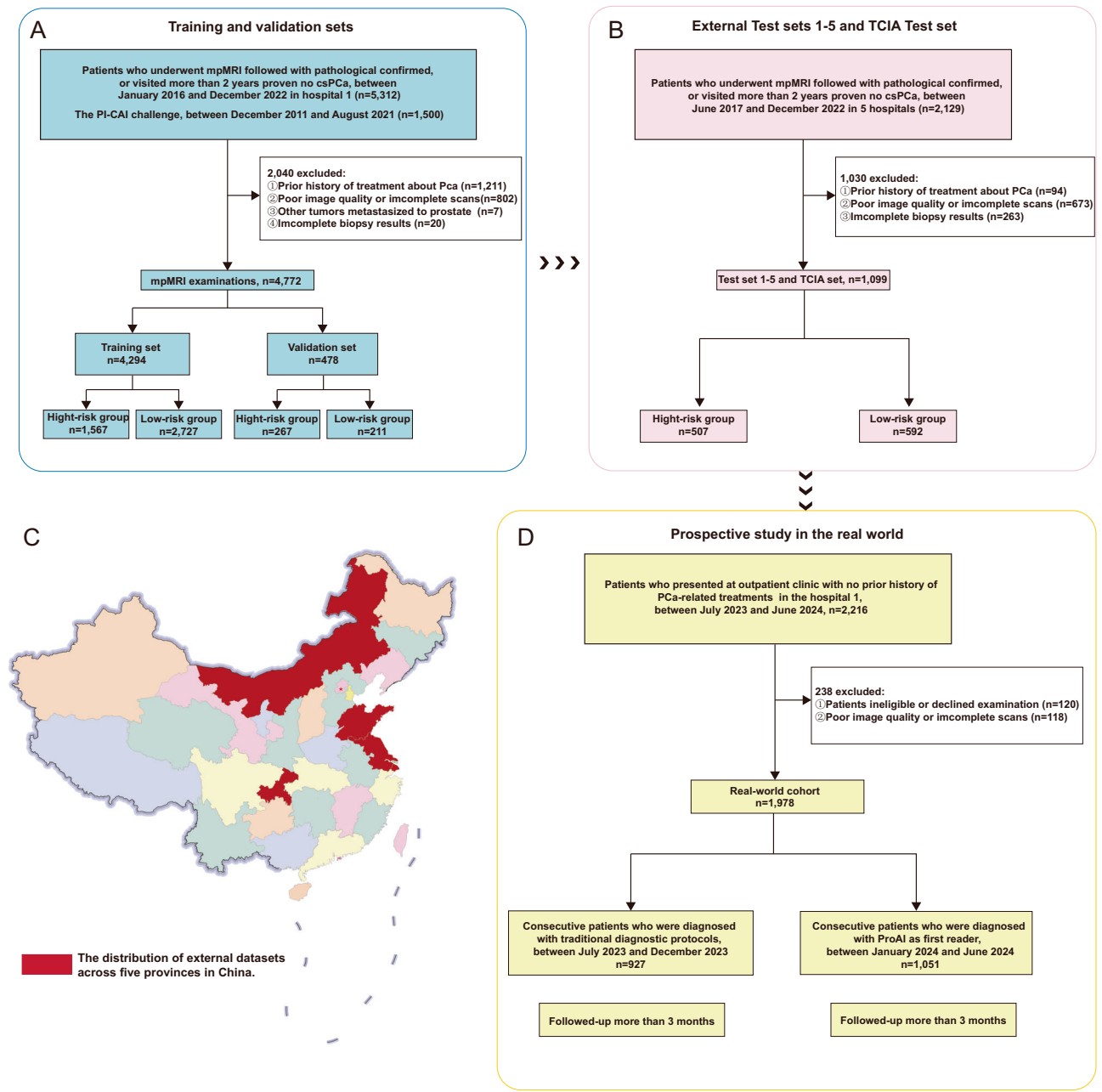

**Fig. 1 | Patient selection.** Flow chart of cohort assembly across (**A**) training and internal validation, **B** external tests (1–5) and TCIA, **D** the prospective real-world periods, and (**C**) geographic distribution of external test cohort patients across Chinese provinces (red-highlighted). Boxes indicate key inclusion/exclusion criteria and final numbers per cohort. Source data are provided as a Source Data file.

reduced resolution. Adhering to PI-RADS technical standards likewise benefits AI stability. ProAI's resilience to protocol variations, mirroring human radiologist patterns. For T2WI resolution, ProAI showed minimal performance variation between high and medium resolution (only specificity $P = 0.03$, NPV $P = 0.01$ differed), while low-resolution imaging affected all metrics ($P < 0.05$). PI-RADS demonstrated similar patterns with significant AUC reduction only in high-to-low resolution comparisons ($P = 0.03$). DWI analysis showed comparable resolution-dependent effects for both ProAI and PI-RADS, confirming that imaging quality impacts AI performance similarly to human interpretation (Supplementary Fig. 5, Supplementary Tables 6–8).

Magnetic field strength analysis revealed modest performance advantages at 3.0 T (AUC 0.92) versus 1.5 T (AUC 0.89), though this difference was not statistically significant ($P = 0.07$) (Supplementary Table 9).

**PI-RADS performance benchmark.** Standard PI-RADS scoring achieved patient-level AUCs of 0.90 (training), 0.85 (validation), and 0.93 (external tests 1–5), with TCIA dataset performance of 0.84 (95% CI: 0.80–0.88) (Table 2, Fig. 3G–J, Supplementary Fig. 2). Inter-observer consistency analysis is detailed in Supplementary Note 4 and Supplementary Fig. 6. DeLong testing revealed significant AUC differences between ProAI and PI-RADS in the training set ($P < 0.001$), while validation and test sets showed comparable performance. At the lesion level, PI-RADS demonstrated superior performance in the validation set ($P < 0.001$) but comparable performance in external tests (Supplementary Table 5).

**Clinical integration success through MRMC validation.** In the AI-assisted condition, only reader-endorsed AI candidates were considered; unendorsed candidates were ignored and did not contribute

**Table 1 | Characteristics of the training, internal validation, external validation, and prospective datasets**

| | Training Set | Validation Set | Test Set. 1 | Test Set. 2 | Test Set. 3 | Test Set. 4 | Test Set. 5 | Test Set. TCIA | Real-world Set (before ProAI) | Real-world Set (after ProAI) | P-value[b] |
|---|---|---|---|---|---|---|---|---|---|---|---|
| Cases | 4,294 | 478 | 246 | 156 | 41 | 51 | 345 | 260 | 927 | 1,051 | |
| Age (yr), (Mean ± SD) | 64.3 ± 10.7 | 68.5 ± 8.1 | 70.0 ± 9.2 | 68.9 ± 8.6 | 68.8 ± 8.5 | 68.1 ± 8.2 | 66.9 ± 8.4 | 65.7 ± 7.5 | 66.4 ± 8.1 | 66.2 ± 9.3 | 0.52[c] |
| NonPCa (%) | 1272(29.6) | 189(39.5) | 68(27.6) | 48(30.8) | 14(34.1) | 22(43.1) | 131(40.0) | 67(25.7) | 165(17.8) | 199(18.9) | 0.83[d] |
| NcsPCa (%) | 292(6.8) | 22(4.6) | 23(9.3) | 6(3.8) | 4(9.8) | 6(11.8) | 27(7.8) | 41(15.8) | 18(1.9) | 22(2.1) | |
| CsPCa (%) | 1567(36.5) | 267(55.9) | 119(48.4) | 77(49.4) | 12(29.2) | 19(37.3) | 128(37.1) | 152(58.4) | 206(22.2) | 241(22.9) | |
| MRBx | 547(12.7) | 0(0) | 0(0) | 0(0) | 0(0) | 0(0) | 0(0) | 0(0) | 0(0) | 0(0) | |
| SysBx | 856(19.9) | 126(26.4) | 171(69.5) | 120(76.9) | 29(70.7) | 45(88.2) | 278(80.6) | 252(96.9) | 102(11.0) | 138(13.1) | |
| SysBx+MRBx | 962(22.4) | 240(50.2) | 0(0) | 0(0) | 0(0) | 0(0) | 0(0) | 8(3.1) | 195(21.0) | 237(22.6) | |
| RP | 766(17.8) | 112(23.4) | 39(15.9) | 11(7.1) | 1(2.4) | 2(3.9) | 8(2.3) | 0(0) | 92(9.9) | 87(8.3) | |
| PI-RADS (%) | | | | | | | | | | | <0.01[d] |
| 1 | 63(1.5) | 11(2.3) | 1(0.4) | 1(0.6) | 0(0) | 2(3.9) | 5(1.4) | 0(0) | 49(5.3) | 71(6.8) | |
| 2 | 2109(49.1) | 108(22.6) | 83(33.7) | 32(20.5) | 18(43.9) | 18(35.3) | 164(47.5) | 73(22.3) | 442(47.7) | 601(57.2) | |
| 3 | 803(18.7) | 85(17.8) | 61(24.8) | 51(32.7) | 6(14.6) | 12(23.5) | 43(12.5) | 85(25.9) | 114(12.3) | 81(7.7) | |
| 4 | 795(18.5) | 133(27.8) | 64(26.0) | 26(16.7) | 4(9.8) | 9(17.6) | 35(10.1) | 111(33.8) | 189(20.4) | 129(12.3) | |
| 5 | 524(12.2) | 141(29.5) | 37(15.0) | 46(29.5) | 13(31.7) | 10(19.6) | 98(28.4) | 59(18.0) | 133(14.3) | 169(16.1) | |
| No. of lesions (%)[a] | | | | | | | | | | | |
| 0 | 154(3.6) | 9(1.84) | 22(8.9) | 10(6.4) | 1(2.4) | 1(1.9) | 31(8.9) | 6(2.3) | NA | 24(2) | |
| 1 | 2872(66.9) | 340(71.1) | 177(71.9) | 107(68.5) | 34(82.9) | 34(66.6) | 242(70.1) | 156(60) | NA | 675(64) | |
| 2 | 973(22.7) | 104(21.7) | 37(15) | 22(14.1) | 2(4.8) | 13(25.4) | 56(16.2) | 61(23.4) | NA | 254(24) | |
| 3 | 254(5.9) | 23(4.8) | 7(2.8) | 9(5.7) | 2(4.8) | 1(1.9) | 12(3.4) | 30(11.5) | NA | 83(8) | |
| 4 | 33(0.8) | 1(0.2) | 2(0.8) | 6(3.8) | 1(2.4) | 2(3.9) | 3(0.8) | 6(2.3) | NA | 14(1) | |
| ≥5 | 8(0.2) | 1(0.2) | 1(0.4) | 2(1.2) | 1(2.4) | 0(0) | 1(0.2) | 1(0.3) | NA | 1(0.01) | |
| No. of patients with PSA (%) | 3231(75.2) | 432(90.4) | 230(93.5) | 154(98.7) | 27(65.9) | 50(98.0) | 313(90.7) | 100(100) | 927(100) | 1,051 (100) | |
| PSA (ng/ml), median (IQR) | 9.3 (6.2, 14.1) | 9.9 (7.0, 19.7) | 10.9 (7.2, 17.0) | 12.5 (6.5, 24.8) | 11.7 (6.1, 20.8) | 12.5 (7.0, 24.0) | 12.1 (6.9, 31.4) | 7.6 (5.2, 11.8) | 9.5(6.4,11.5) | 8.7(6.3,12.6) | 0.28[d] |

*ISUP International Society of Urology Pathology, non-PCa non-prostate cancer, ncsPCa non-clinically significant prostate cancer (ISUP = 1), csPCa Clinically significant prostate cancer (ISUP > 1), IQR interquartile range, MRBx MRI-guided prostate biopsy, SysBx systematic prostate biopsy, SD standard deviation, RP radical prostatectomy, PSA prostate specific antigen; [a]All lesions were generated by AI model. [b]P-value was compared between group before and after ProAI implementation. [c]P-value calculated from t-tests. [d]P-value calculated from Mann-Whitney U test.*

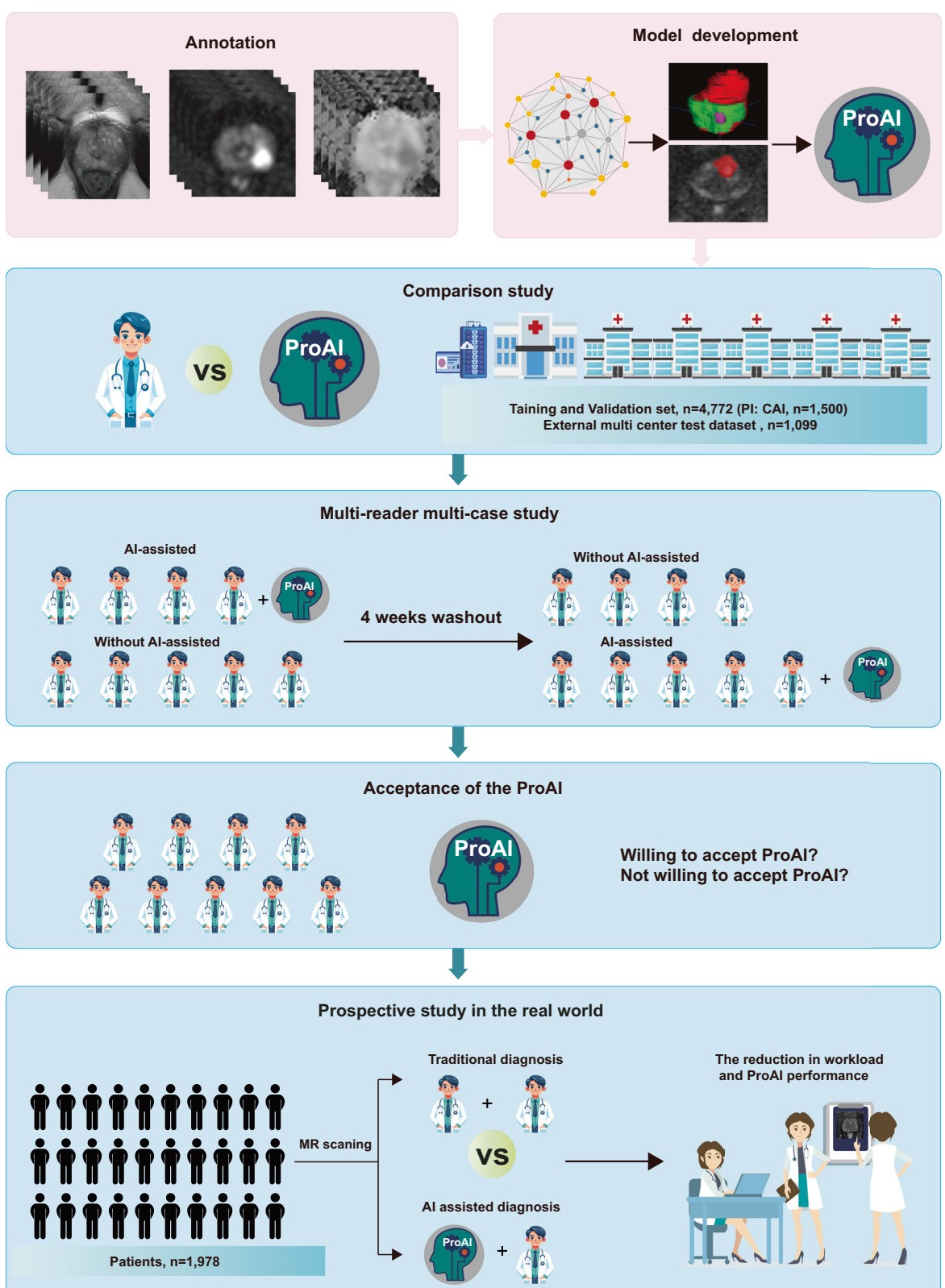

**Fig. 2 | Development and validation framework.** Flow chart of the four-stage pipeline for ProAI: model development, internal validation, external testing (tests 1–5 and TCIA), and prospective clinical implementation. Arrows indicate data flow, outputs (segmentation and risk score), and evaluation endpoints. Source data are provided as a Source Data file.

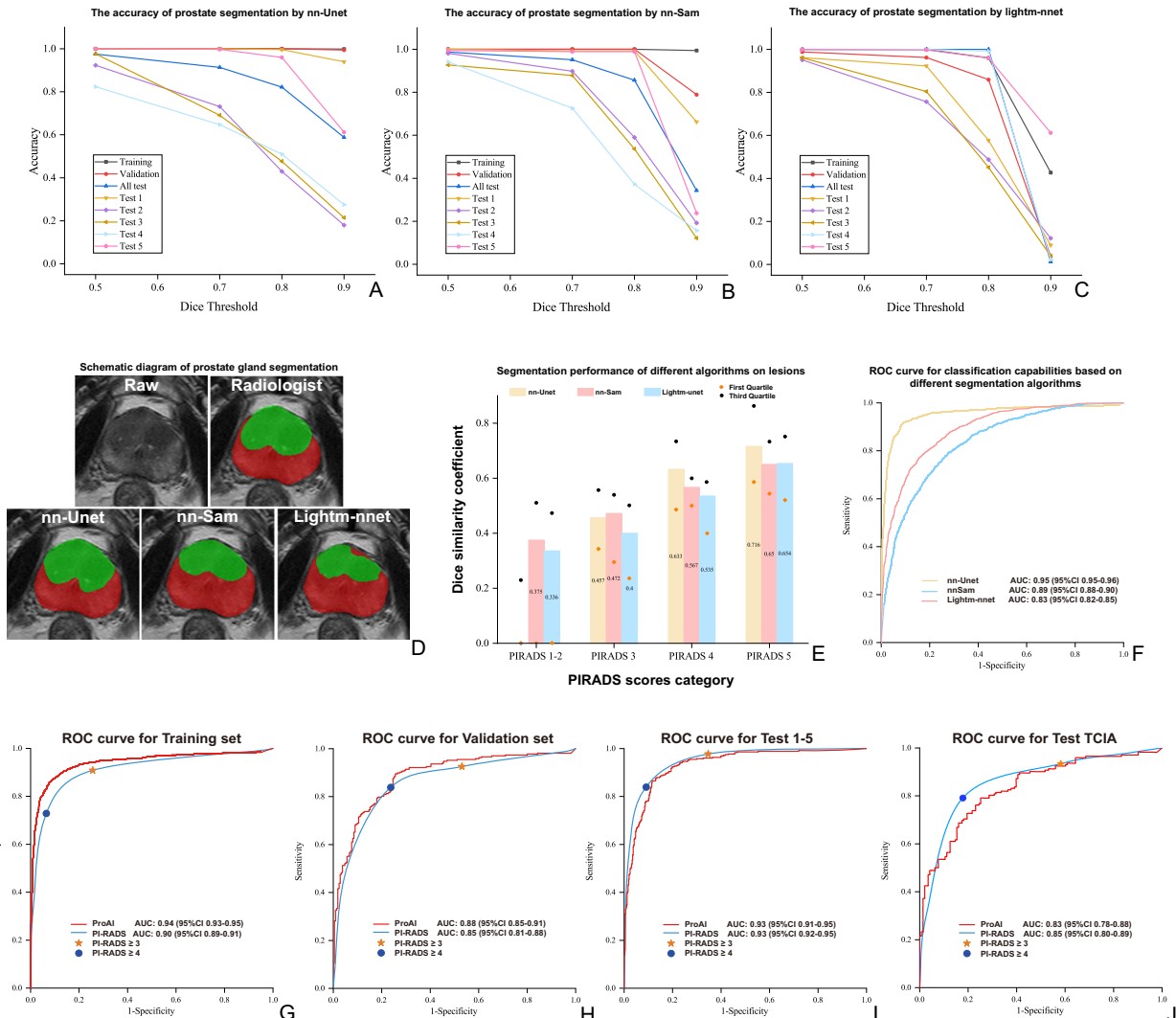

**Fig. 3 | Segmentation accuracy and diagnostic performance across models. A–C** Prostate-gland Dice across nnUNet, nnSAM and LightM-UNet. **D** Schematic of gland segmentation. **E** Lesion-level Dice across models. **F** ROC curves for lesion classification. **G–J** Patient-level ROC curves for PI-RADS and ProAI across datasets; AUCs were compared using DeLong's test. All statistical tests were two-sided with $P < 0.05$ considered significant unless stated. ROC receiver operating characteristic, AUC area under the ROC curve, TCIA The Cancer Imaging Archive. Source data are provided as a Source Data file.

to patient-level metrics. The multi-reader multi-case study involving nine clinicians across 250 cases demonstrated significant diagnostic enhancement with ProAI assistance. Overall AUC improved from 0.80 (95% CI: 0.74–0.86) to 0.86 (95% CI: 0.82–0.90) ($P < 0.01$). Stratified analysis revealed differential benefits: senior radiologists showed minimal improvement ($P = 0.42$), while general radiologists and urologists achieved significant enhancement (both $P < 0.01$). Diagnostic consistency among physicians improved significantly with ProAI assistance ($P < 0.01$), except for senior radiologists ($P = 0.91$).

Workflow efficiency gains were substantial, with average interpretation time reduced from $72.7 \pm 23.5$ seconds to $48.7 \pm 10.0$ seconds ($P < 0.01$) for all readers except the most senior. These timing measurements reflect active diagnostic processes, with ProAI preprocessing occurring parallel to clinical workflow. Detailed performance changes are shown in Fig. 4, Supplementary Figs. 7–9, and Supplementary Tables 10–12.

**Strategic clinical integration patterns.** Clinical integration assessment revealed sophisticated utilization patterns beyond simple acceptance. AI consultation occurred in 91.13% (2163/2250) of cases (Fig. 4I), with strategic usage patterns correlating inversely with physician confidence ($r = −0.64$, $P < 0.001$). Non-consultation cases (8.87%) were predominantly high-confidence scenarios (mean rating 4.7/5), particularly among senior radiologists.

Case difficulty significantly influenced consultation patterns: 97.2% for low-conspicuity lesions, 93.6% for moderate difficulty, and 86.4% for high-conspicuity cases ($P < 0.01$). Reader stratification confirmed targeted benefits: general radiologists improved from AUC 0.76 to 0.85 ($P < 0.01$), urologists from 0.74 to 0.84 ($P < 0.01$), while senior radiologists remained stable (0.89 to 0.90, $P = 0.42$) (Fig. 4G). Technology Acceptance Model questionnaire results showed high perceived usefulness (4.3/5), moderate integration ease (3.9/5), and strong implementation intention (4.5/5) (Supplementary Table 13).

**Real-world implementation impact**
Prospective implementation demonstrated tangible clinical benefits. Disagreement rates on recall decisions decreased significantly from 9.92% (92/927) to 7.23% (76/1051) ($P < 0.01$). For high-risk cases, senior radiologist-ProAI consistency reached 94.67% (355/375), requiring

**Table 2 | Comparison of patient-level diagnostic performance between the AI model and PI-RADS in all datasets**

| | AUC (95%CI) | Z value | P value[a] | Sensitivity | Specificity | Accuracy | PPV | NPV |
|---|---|---|---|---|---|---|---|---|
| **AI model** | | | | | | | | |
| Training set | 0.94(0.93–0.95) | - | - | 0.86(1345/1567) | 0.92(2519/2727) | 0.90(3864/4294) | 0.87(1345/1553) | 0.92(2519/2741) |
| Validation set | 0.88(0.85–0.91) | - | - | 0.90(239/267) | 0.74(157/211) | 0.83(396/478) | 0.82(239/293) | 0.85(157/185) |
| Test set 1-5 | 0.93(0.91–0.95) | - | - | 0.90(319/355) | 0.87(419/484) | 0.88(738/839) | 0.83(319/384) | 0.92(419/455) |
| Test set 1 | 0.93(0.90–0.96) | - | - | 0.93(111/119) | 0.81(103/127) | 0.87(214/246) | 0.82(111/135) | 0.93(103/111) |
| Test set 2 | 0.86(0.81–0.92) | - | - | 0.83(64/77) | 0.76(60/79) | 0.79(124/156) | 0.77(64/83) | 0.82(60/73) |
| Test set 3 | 0.90(0.79–1.0) | - | - | 0.83(10/12) | 0.86(25/29) | 0.85(35/41) | 0.71(10/14) | 0.93(25/27) |
| Test set 4 | 0.92(0.84–0.99) | - | - | 0.79(15/19) | 0.91(29/32) | 0.86(44/51) | 0.83(15/18) | 0.88(29/33) |
| Test set 5 | 0.96(0.94–0.99) | - | - | 0.93(119/128) | 0.93(202/217) | 0.93(321/345) | 0.89(119/134) | 0.96(202/211) |
| Test set TCIA | 0.83(0.78–0.88) | | | 0.75(109/146) | 0.76(81/106) | 0.75(190/252) | 0.81(109/134) | 0.69(81/118) |
| **PI-RADS** | | | | | | | | |
| Training set | 0.90(0.89–0.91) | 8.674 | <0.001 | 0.91(1423/1567) | 0.74(2028/2727) | 0.80(3451/4294) | 0.67(1423/2122) | 0.93(2028/2172) |
| Validation set | 0.85(0.81–0.88) | 1.878 | 0.060 | 0.93(247/267) | 0.47(99/211) | 0.72(346/478) | 0.69(247/359) | 0.83(99/119) |
| Test set 1-5 | 0.93(0.92--0.95) | 0.274 | 0.784 | 0.98(347/355) | 0.65(316/484) | 0.79(663/839) | 0.67(347/515) | 0.98(316/324) |
| Test set 1 | 0.91(0.88–0.95) | 0.827 | 0.408 | 0.97(116/119) | 0.64(81/127) | 0.80(197/246) | 0.72(116/162) | 0.96(81/84) |
| Test set 2 | 0.90(0.85–0.94) | 1.343 | 0.179 | 0.99(76/77) | 0.41(32/79) | 0.69(108/156) | 0.62(76/123) | 0.97(32/33) |
| Test set 3 | 0.93(0.87–1.0) | 0.662 | 0.508 | 1.00(12/12) | 0.62(18/29) | 0.73(30/41) | 0.52(12/23) | 1.00(18/18) |
| Test set 4 | 0.93(0.86–1.0) | 0.403 | 0.687 | 1.00(19/19) | 0.62(20/32) | 0.76(39/51) | 0.61(19/31) | 1.00(20/20) |
| Test set 5 | 0.96(0.94–0.98) | 0.003 | 0.998 | 0.97(124/128) | 0.76(165/217) | 0.84(289/345) | 0.70(124/176) | 0.98(165/169) |
| Test set TCIA | 0.85(0.80–0.89) | 1.153 | 0.249 | 0.94(143/152) | 0.42(45/108) | 0.72(188/260) | 0.69(143/206) | 0.83(45/54) |

*AI* artificial intelligence, *AUC* area under the curve, *CI* confidence interval, *PI-RADS* prostate imaging reporting and data system, *PPV* positive predictive value, *NPV* negative predictive value.
[a]*P*-values calculated from the DeLong test between AI model and PI-RADS scores.
AUCs are calculated from continuous scores. For the AI model, sensitivity/specificity/accuracy/PPV/NPV in each test set are computed at a per-dataset threshold selected by maximizing Youden's J; for PI-RADS, a fixed threshold of ≥3 is used. The pooled 'Test set 1-5' metrics are micro-averaged by summing TP/FP/TN/FN across the datasets at their respective thresholds.

additional review in only 5.33% (20 cases). Low-risk cases maintained 91.72% consistency (620/676) with double-reading protocols, necessitating further review in 8.28% (56 cases).

Qualitative disagreement analysis revealed specific failure patterns. In high-risk cases, inconsistencies (5.33%) stemmed from subtle DWI findings (45%), complex anatomy (30%), technical factors (15%), and atypical presentations (10%). Among 11 biopsied cases, 3 (27.3%) confirmed csPCa, yielding a clinically significant false negative rate of 0.8% (3/375) (Supplementary Table 14). Low-risk inconsistencies (8.28%) involved transitional zone lesions (39.3%), borderline PI-RADS 3/4 cases (33.9%), prostatitis mimics (14.3%), hemorrhagic changes (8.9%), and technical limitations (3.6%). Of 42 biopsied cases, 17 (40.5%) confirmed csPCa, indicating ProAI's ability to detect radiologist-missed cases (Supplementary Table 15).

The workflow optimization enabled 64.32% (676/1051) of screenings to be single-read, achieving a 32.16% workload reduction. Three-month follow-up showed comparable pathological confirmation rates before (42.96%, 389/927) and after (43.96%, 462/1051) implementation, with consistent csPCa detection rates (52.96% vs. 52.16%).

### Diagnostic performance enhancement

PI-RADS scores were treated as continuous variables for AUC analysis, while a dichotomized cutoff of ≥3 was used for sensitivity/specificity. Post-implementation diagnostic metrics demonstrated significant improvements: AUC increased to 0.94 (95% CI: 0.90−0.95), sensitivity to 0.97 (95% CI: 0.94−0.99), and specificity to 0.88 (95% CI: 0.83−0.92). which was superior to the pre-implementation period (AUC 0.90; sensitivity 0.92; specificity 0.75), were statistically significant (*P* < 0.05; Fig. 5C). Representative cases are illustrated in Fig. 6.

Implementation practicality metrics confirmed seamless integration: radiologist proficiency achieved within 1.5 days (range: 1-3), PACS-to-ProAI access time of 2.7 ± 0.9 seconds, and 100% successful workflow incorporation. Post-implementation surveys showed 87% (20/23) of radiologists rating integration as 'good' or 'excellent', emphasizing minimal workflow disruption (Supplementary Fig. 10).

## Discussion

Prostate cancer remains the second most common malignancy in men globally, yet accurate non-invasive detection of clinically significant disease continues to challenge current diagnostic paradigms[16]. While multiparametric MRI with PI-RADS scoring represents the current standard of care, substantial limitations persist, including marked inter-observer variability (κ = 0.577−0.683) and suboptimal performance at intermediate risk categories[6,17]. These diagnostic uncertainties necessitate invasive biopsies with associated patient morbidity and healthcare costs, highlighting the urgent need for more reliable, objective diagnostic approaches.

Our study introduces ProAI, a fully automated deep learning system that addresses fundamental limitations of current diagnostic approaches through several key innovations. The segmentation-before-classification architecture provides explicit lesion localization with median centroid accuracy of 2.8 mm (IQR: 1.5−4.3 mm), enabling potential integration with targeted biopsy workflows. Critically, our mixed training strategy using both manually annotated and AI-segmented lesions creates robustness to real-world segmentation imperfections, maintaining AUC > 0.89 even with substantial segmentation errors (DSC ≤ 0.5). The comparative evaluation of segmentation architectures (nnUNet, nn-SAM, LightM-UNet) establishes nnUNet's current superiority for prostate applications[18], while identifying pathways for future transformer-based enhancements[19].

The scale and diversity of our validation represents a significant advancement over previous studies. With 7849 cases across 34 MR scanners of 21 models from 6 medical centers plus the ethnically diverse PI-CAI[14] and TCIA[20] datasets, ProAI demonstrates consistent performance across varied populations and acquisition protocols.

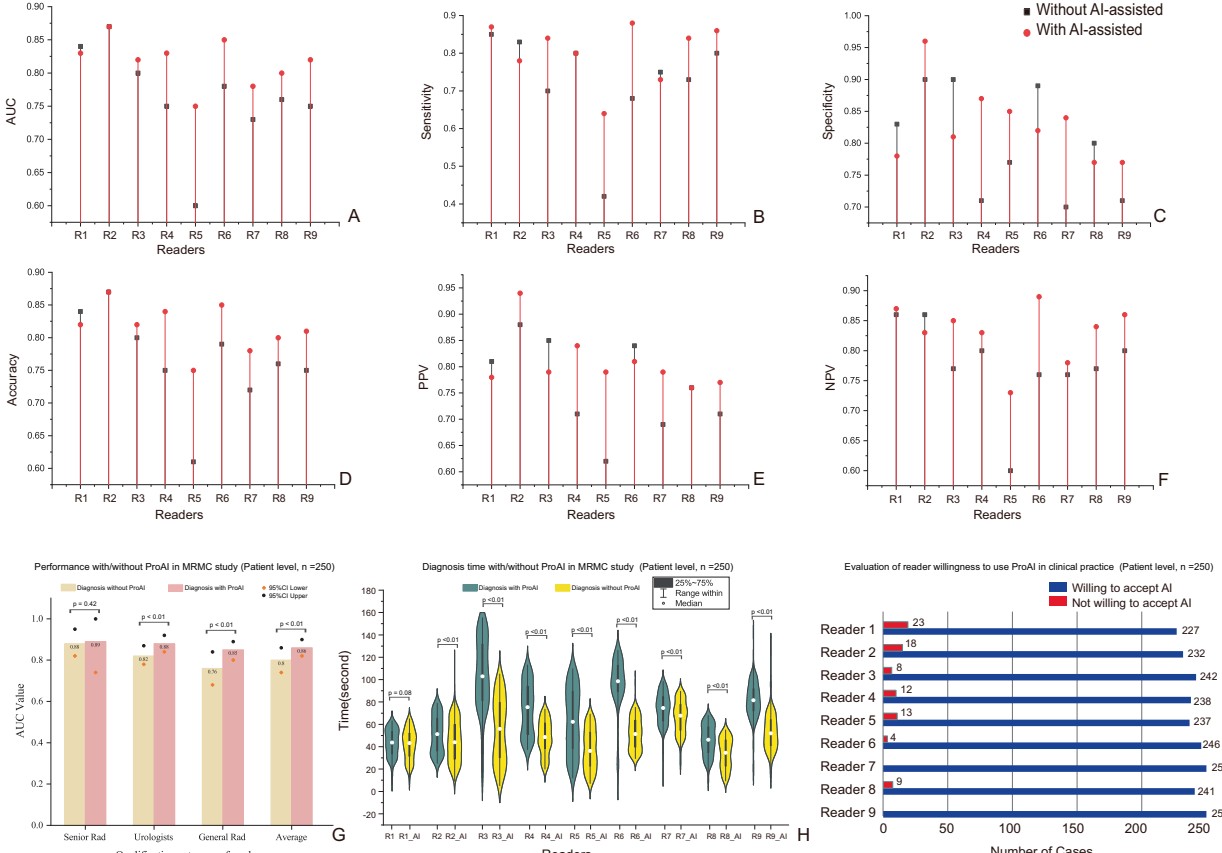

**Fig. 4 | Multi-reader multi-case evaluation and workflow impact. A–F** Per-reader performance without (black) and with (red) ProAI for: (**A**) AUC, (**B**) sensitivity, (**C**) specificity, (**D**) accuracy, (**E**) PPV and (**F**) NPV. Nine readers (R1–R9) each interpreted $n = 250$ de-identified cases per condition; points show per-reader estimates and lines link paired conditions. AUCs were computed from ordinal reader scores; other metrics were calculated at the prespecified diagnostic threshold used in the reading study. Group-level comparisons were performed within the OR–MRMC framework (two-sided) and are reported in the text and Supplementary Tables. **G** Experience-stratified AUC for senior radiologists (>10 years prostate MRI), general radiologists (<5 years prostate MRI) and urologists. Bars show group means with 95% CIs; two-sided tests as specified in Methods. **H** Reading time per case with and without ProAI. Violin plots summarise $n = 250$ cases per reader per condition; central line = median; box = IQR (25th–75th percentiles); whiskers = non-outlier range; overlaid dots indicate mean ± SD. Within-reader differences were assessed with a two-sided paired t-test (no multiplicity adjustment). Mean time decreased from 72.7 ± 23.5 s to 48.7 ± 10.0 s. Timing reflects active interpretation only and excludes AI preprocessing, which runs in parallel. **I** Clinical integration: consultation counts per reader across 250 cases; overall ProAI consultation rate 91.13%. Usage patterns reflect case complexity and reader confidence. All statistical tests were two-sided, and $P < 0.05$ was considered significant unless stated. AUC area under the ROC curve, PPV positive predictive value, NPV negative predictive value, R reader, AI ProAI software. Source data are provided as a Source Data file.

Unlike previous large-scale studies that predominantly relied on single-scanner datasets[14], our multicenter approach ensures genuine generalizability. The maintained performance across Asian (AUC: 0.86–0.97) and Western populations (AUC: 0.83) validates cross-ethnicity applicability, though slightly reduced performance in the TCIA cohort suggests opportunities for population-specific calibration.

Our systematic evaluation of clinical integration represents a paradigm shift from technical validation to real-world implementation. The multi-reader multi-case study ($n = 9$ readers, 250 cases) provides robust evidence that AI assistance significantly enhances diagnostic performance (AUC: $0.80 \rightarrow 0.86$, $P < 0.01$), with greatest benefit observed among general radiologists and urologists—precisely the populations most likely to benefit from AI support in resource-limited settings[21]. The strategic AI consultation patterns (91.13% overall usage, inverse correlation with physician confidence r = −0.64) demonstrate sophisticated clinical integration rather than simple tool adoption.

The prospective implementation study establishes concrete evidence for AI's healthcare value proposition. The 32.16% reduction in radiologist workload while maintaining diagnostic excellence (AUC: 0.92) demonstrates tangible efficiency gains crucial for healthcare sustainability[22]. The innovative workflow integration—where AI

determines single versus double reading requirements—represents a approach to resource optimization that could be broadly applicable across healthcare systems. During prospective deployment, the rate of clinically meaningful human–AI disagreements—defined as discordance that would change management stratification—was 3.5% (69/1978 cases; Fig. 5B), supporting the reliability of the collaborative workflow.

Our findings have broader implications beyond prostate cancer screening. The demonstrated ability of AI to elevate less experienced practitioners to near-expert performance levels addresses critical healthcare disparities, particularly relevant for regions with limited subspecialty expertise[23]. The 40.5% detection rate of radiologist-missed csPCa cases in high-risk AI predictions suggests complementary rather than competitive human-AI interaction, supporting augmentation rather than replacement paradigms[24].

Several limitations warrant consideration. The mixed pathological standards (biopsy versus prostatectomy) may introduce verification bias, though this reflects real-world clinical practice. While our TCIA validation demonstrates applicability in more diverse populations, more comprehensive validation across explicitly categorized demographic groups would strengthen generalizability claims. The implicit size processing through 3D convolutions, while effective, could benefit

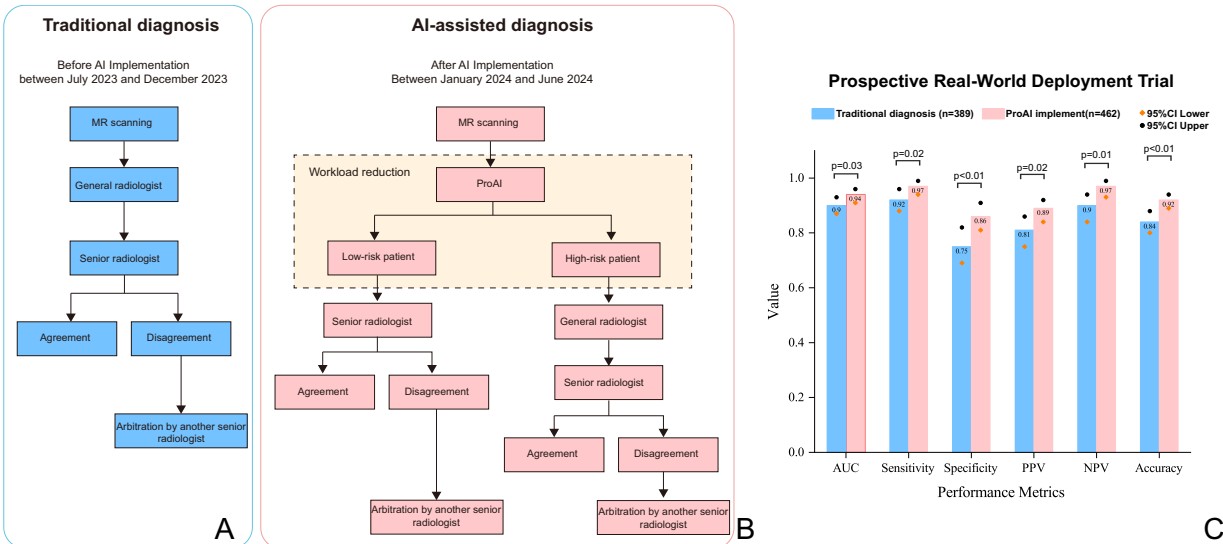

**Fig. 5 | Diagnostic workflow and performance: before versus after AI implementation. A**, **B** The prospective design flowchart and (**C**) result comparison. All statistical tests were two-sided, with *P* < 0.05 indicative of a statistically significant difference. AUC, AUCs; PPV, positive predictive value; NPV, negative predictive value; AI, ProAI software. Source data are provided as a Source Data file. Note: **B** AI candidates required reader endorsement to enter the decision; unendorsed candidates were not counted as positives. **C** Performance metrics: AUC was derived from continuous scores. Sensitivity/specificity utilized a fixed PI-RADS threshold (≥3).

from explicit dimensional parameter integration for borderline cases. Future architectural developments should explore hybrid segmentation-classification approaches and uncertainty quantification to further enhance clinical reliability.

Based on our findings, we propose a structured approach for broader clinical implementation: (1) institutional pilot programs with existing PACS integration pathways, (2) population-specific calibration studies for diverse demographic groups, (3) health economic evaluations to quantify cost-effectiveness[25], and (4) regulatory pathway development for clinical approval. The demonstrated physician acceptance and workflow compatibility suggest that technical barriers to adoption are surmountable, with primary challenges likely residing in regulatory and reimbursement domains.

In conclusion, ProAI represents a clinically validated AI system that addresses fundamental limitations of current prostate cancer diagnostics while demonstrating concrete healthcare benefits. The combination of technical innovation, rigorous validation, and demonstrated real-world impact establishes a framework for successful AI integration in cancer screening, with implications extending beyond prostate cancer to broader oncologic applications. Our findings support the clinical implementation of AI-assisted diagnosis as a mean to enhance diagnostic accuracy, reduce healthcare costs, and improve patient outcomes in prostate cancer management.

## Methods
### Study design and ethical considerations
This study adhered to the Declaration of Helsinki and obtained approval from the ethics committees of Hospital 1 (The Changhai Hospital) (approval CHEC-Y2024-015). This multicenter study adhered to STARD[26] and TRIPOD[27] guidelines for diagnostic and prognostic research, incorporating both retrospective development phases and prospective clinical validation.

The retrospective component collected consecutive patient data from six hospitals (January 2016-December 2022) and PI-CAI[14] (December 2011-August 2021) and TCIA[20] (January 2006-February 2011). Ethical approval was obtained from Hospital 1 Ethics Committee (CHEC-Y2024-015), with retrospective data exempted from informed consent requirements per institutional guidelines. The prospective

validation phase enrolled patients with suspected prostate cancer at Hospital 1 (July 2023-June 2024), with all participants providing written informed consent.

The study employed a systematic four-phase validation framework (Fig. 2, Supplementary Fig. 1): (1) AI development and technical validation across internal and external datasets, including performance comparison with radiologist PI-RADS assessments using Hospital 1 and public dataset splits for training/validation, with additional external validation across five hospitals (tests 1-5) and the TCIA dataset using self-reported ethnicity for analysis; (2) multi-reader multi-case (MRMC) evaluation using 250 randomly selected cases from internal validation and test 1 to assess AI-assisted diagnostic enhancement; (3) clinical integration assessment examining physician utilization patterns and acceptance using identical MRMC datasets; and (4) prospective real-world implementation evaluating clinical decision-making impact and workflow efficiency at Hospital 1.

### Patient selection and clinical criteria
Inclusion criteria comprised: (1) patients undergoing prostate biopsy or radical prostatectomy with histopathological confirmation, and (2) multiparametric MRI performed within 8 weeks prior to tissue sampling. Exclusion criteria included: (1) prior androgen deprivation therapy, focal therapy, radiotherapy, or chemotherapy; (2) suboptimal MRI quality or incomplete sequences; (3) non-prostatic malignancies (bladder cancer, sarcomas) with prostatic involvement; and (4) incomplete histopathological results.

### Sample size determination
Statistical power calculations utilized PASS software (v21.0.3, NCSS Statistical Software, USA) for comparison studies (phases 1 and 4) and RJafroc package in R (v4.6.1, R Foundation for Statistical Computing, USA) for MRMC analysis (phase 2). Detailed sample size estimations are provided in Supplementary Note 5. Clinical data extraction followed standardized protocols from institutional medical record systems (Supplementary Note 6).

### Advanced MRI acquisition and standardization
All participants underwent standardized multiparametric MRI including T2-weighted imaging (T2WI), diffusion-weighted imaging (DWI),

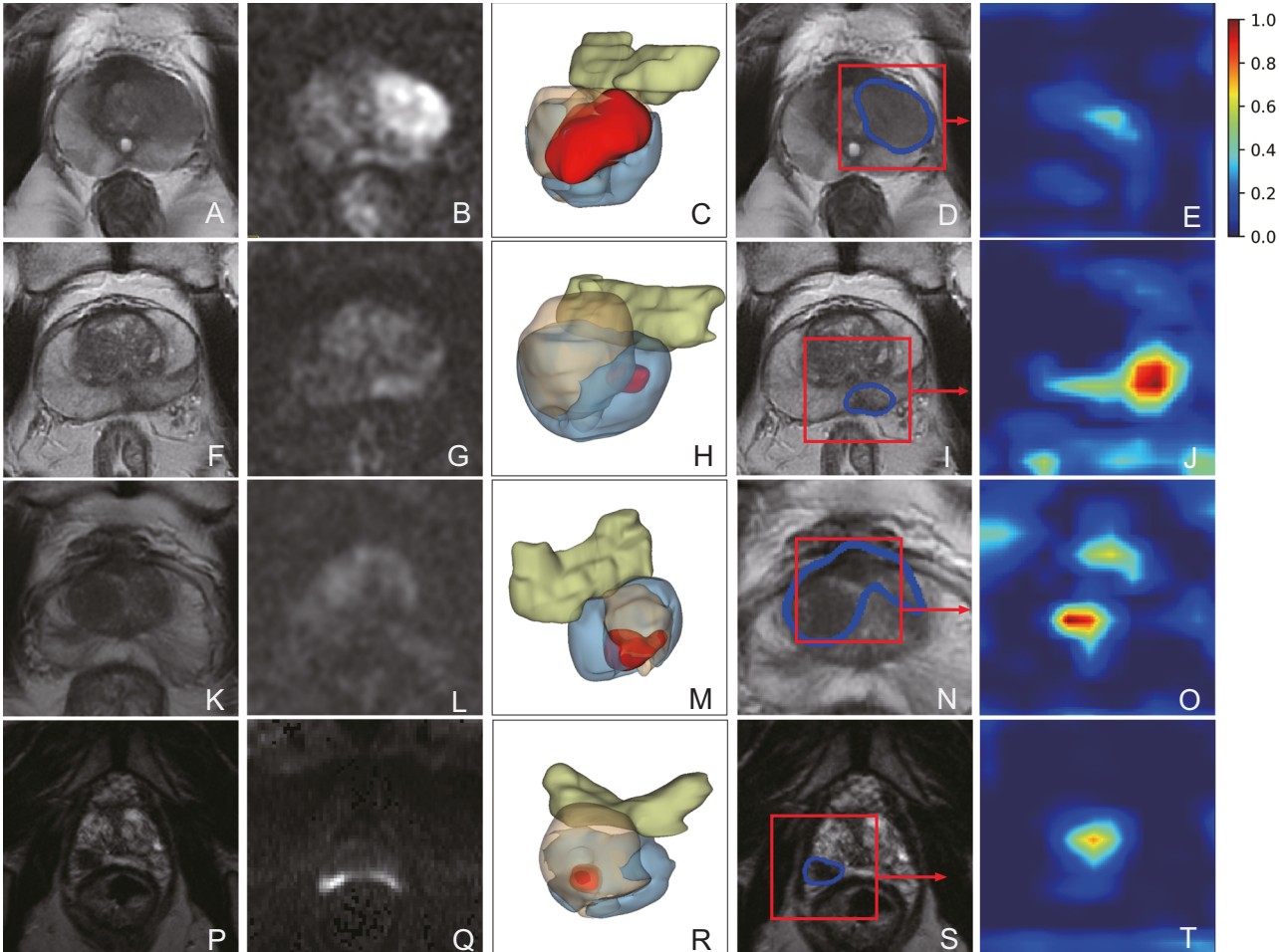

**Fig. 6 | Representative cases across diverse clinical scenarios. Case A (A–E):** 51-year-old man, PSA 13.3 ng/mL, left peripheral-zone lesion, PI-RADS 3. T2W and high-b DWI show a focal finding; ProAI delineates prostate and lesion and produces a low-probability heatmap (0.114). Biopsy: chronic inflammation; lesion resolved after targeted therapy. **Case B (F–J):** 66-year-old man, PSA 6.5 ng/mL, subtle peripheral-zone lesion initially PI-RADS 3 and overlooked on routine reading. ProAI identifies and segments the lesion; heatmap probability 0.922. Targeted biopsy and radical prostatectomy (RP) confirm csPCa (ISUP Grade Group 2). **Case C (K–O):** 70-year-old man, PSA 8.3 ng/mL, transition-zone lesion PI-RADS 2, a challenging differentiation from benign hyperplasia. ProAI localises the lesion; heatmap probability 0.964. RP confirms csPCa (ISUP Grade Group 2). **Case D (P–T):** 64-year-old man, PSA 3.9 ng/mL, right peripheral-zone lesion PI-RADS 4. Despite low spatial resolution on T2W and motion artefact on DWI, ProAI maintains robust segmentation; heatmap probability 0.856. RP confirms high-grade csPCa (ISUP Grade Group 5). PSA prostate-specific antigen, PI-RADS Prostate Imaging Reporting and Data System, T2W T2-weighted, DWI diffusion-weighted imaging, PZ/TZ peripheral/transition zone, RP radical prostatectomy, csPCa clinically significant prostate cancer, ISUP International Society of Urological Pathology.

and dynamic contrast-enhanced (DCE) sequences. Comprehensive scanning protocols and equipment specifications across six hospitals and the public dataset are detailed in Supplementary Table 16 and Supplementary Note 6, ensuring technical heterogeneity representative of real-world clinical practice.

**Expert image interpretation and quality assurance**
Two blinded genitourinary radiologists (6- and 15-years' experience) independently re-evaluated all MRI examinations from six datasets. Discrepancies underwent structured consensus review involving both primary readers and a senior radiologist (25 years' experience) through systematic discussion of discrepant findings. Final determinations required either unanimous agreement or majority consensus when complete agreement was unattainable. All interpretations followed PI-RADS v2.1 guidelines[4], incorporating T2WI, DWI, ADC maps, and DCE sequences. Both lesion-level and patient-level diagnoses were assigned, with patient-level classification determined by the highest-grade lesion in cases with multiple abnormalities.

**Histopathological analysis and correlation**
Histopathological evaluation was performed by two experienced urological pathologists (10- and 15-years' experience), with discrepancies resolved by a third pathologist (>20 years' experience). Both biopsy and radical prostatectomy specimens were analyzed, with surgical pathology considered definitive when both were available. Classifications followed International Society of Urological Pathology (ISUP) guidelines[15] using Gleason scoring. Patients were stratified as non-clinically significant prostate cancer (ncsPCa; Gleason grade group 1) or clinically significant prostate cancer (csPCa; Gleason grade groups 2-5), then dichotomized into low-risk (negative pathology/ncsPCa) and high-risk (csPCa) categories.

**Robust MRI-pathology correlation protocol**
Standardized correlation protocols were implemented by a multidisciplinary team including an experienced genitourinary radiologist (Y.B., 15 years' experience) and urological pathologist (H.J., 12 years' experience). Correlation approaches varied by specimen type (radical prostatectomy, combined systematic/targeted biopsies, or systematic

biopsy alone), with specialized protocols for challenging cases and quality control measures ensuring robust correlation (inter-observer $\kappa = 0.82$). Detailed methodology is provided in Supplementary Note 7.

## ProAI architecture development and innovation

**Lesion annotation and preprocessing framework.** Expert radiologists (8- and 9-years' diagnostic experience) performed precise lesion annotations using open-source ITK-SNAP software (Version 3.8.0, http://www.itksnap.org, 2020). The preprocessing pipeline incorporated registration, cropping, and normalization algorithms optimized for multicenter data heterogeneity.

## Segmentation architecture evaluation and selection

**Pipeline selection and architecture screening.** We adopted a segmentation-then-classification (STC) pipeline. Within this paradigm, we conducted a systematic evaluation of segmentation architectures—including nnUNet[18], nn-SAM[28], and LightM-UNet[29]—and selected the final configuration based on (i) lesion localization/segmentation quality and (ii) downstream patient-level csPCa AUC using fixed classification heads. We did not perform a head-to-head cross-paradigm benchmark (STC vs classification-then-segmentation vs joint segmentation-classification); any non-STC trials were preliminary explorations only and are not presented as systematic baselines. This choice aligns with clinical workflow and provides explicit lesion localization for interpretability. Three state-of-the-art segmentation architectures underwent comparative evaluation representing distinct technological approaches: nnUNet (self-configuring convolutional architecture), nn-SAM (hybrid transformer-convolutional design integrating Medical SAM vision transformer structure with nnUNet self-adaptive capabilities), and LightM-UNet (lightweight efficient architecture designed for medical image segmentation). While MedSAM represents significant advancement through Segment Anything Model foundation, its interactive prompt-based optimization differs from our fully automated requirements, necessitating nn-SAM evaluation as a more suitable automated alternative.

## Classification model architecture and robustness enhancement

The classification model processed eight input channels: T2WI, fat-suppressed T2WI (FS-T2WI), ADC, and high b-value DWI sequences (DWI1000, DWI1500, DWI2000, DWI3000), plus lesion mask. To enhance robustness across protocol variations, strategic channel dropout was implemented during training, ensuring either T2WI or FS-T2WI retention (but not both) and random high b-value DWI selection per training iteration. This augmentation strategy improved model generalizability across centers with heterogeneous imaging protocols.

Classification utilized a CNN model based on HarDNet[30] (Harmonic Densely Connected Network), optimized for low memory access costs (MACs) and memory traffic while maintaining DenseNet performance advantages. A mixed-training approach enhanced pipeline robustness: the classifier trained on both (i) manually annotated lesions by experienced radiologists and (ii) segmentation model-identified lesions. This dual-source strategy specifically adapted the classifier to segmentation output characteristics, including potential boundary inaccuracies, ensuring optimal real-world performance. Implementation details are provided in Supplementary Note 8, with the complete development process illustrated in Supplementary Fig. 11.

## Multi-reader multi-case study design

MRMC evaluation utilized 250 randomly selected mpMRI examinations from validation and test 1 datasets in a randomized, open-label comparison design. Nine independent readers from Hospitals 1 and 2 participated, with none involved in patient recruitment or image labeling. Detailed methodology is described in Supplementary Note 9.

## Handling of AI overcalled candidates

At inference, all AI-generated segmentations were displayed as candidate overlays. Readers were instructed to endorse or dismiss each candidate based on clinical plausibility. Overcalled lesions were defined as AI masks with no match to any annotated or pathology-proven target under dual criteria (no spatial overlap above the preset IoU/Dice threshold and no centroid proximity within $d$ mm). Unendorsed AI candidates were excluded from patient-level analyses; patient-level positivity required either a reader-endorsed AI candidate matching the reference standard or a reader-identified non-AI lesion. This policy prevents unendorsed AI candidates from inflating patient-level false positives.

## Clinical integration assessment protocol

Clinical integration evaluation (November 2023) employed identical MRMC datasets with the same nine readers. Multiple measurement approaches assessed integration patterns beyond usage statistics: (1) validated 10-item Technology Acceptance Model (TAM) questionnaire evaluating perceived usefulness, ease of use, and clinical implementation intention (5-point Likert scale); (2) reader confidence ratings (1-5 scale) for each case before AI result viewing decisions, with AI outputs accessible via specialized software interface rather than automatic display; (3) independent case difficulty rating based on lesion conspicuity (high, moderate, low) by two study-independent radiologists, enabling AI consultation pattern analysis across complexity levels. A 6-week washout period between reading sessions and randomized case presentation order minimized recall bias, as detailed in Supplementary Note 9.

## Prospective real-world implementation study

**Conventional workflow baseline.** Standard prostate MRI interpretation involved two independent radiologists (general radiologist for initial assessment, experienced radiologist for review), with disagreements requiring third-radiologist arbitration.

## AI-integrated workflow innovation

ProAI integration streamlined the screening process: automatic MRI upload to the AI platform provided lesion highlighting and initial csPCa diagnosis. Low-risk screenings underwent single senior radiologist review, while high-risk cases received double reading with ProAI decision support. Senior radiologist recalls recommendations triggered additional radiologist arbitration. The primary outcome measured workload reduction through reading volume quantification.

Three-month follow-up assessed diagnostic accuracy using histopathological gold standards (radical prostatectomy, systematic biopsy, or combined targeted/systematic biopsy). Post-implementation performance (January-June 2024) was compared to pre-implementation workflow (July-December 2023). The prospective design is illustrated in Fig. 5A, B.

## Technical implementation and clinical integration architecture

ProAI employed browser/server (B/S) architecture providing web-based DICOM image AI viewer with flexible PACS integration pathways. Direct integration enabled dedicated PACS button redirection to specified examinations with secure encrypted URL parameter transmission. DICOM routing-capable systems utilized ProAI as a network DICOM node for automatic processing and result delivery. Limited integration systems employed floating OCR utility tools capturing user-selected examination numbers for one-click web-based viewer access. These flexible approaches ensured successful implementation across diverse hospital IT environments with minimal workflow disruption. The specific operation methods of the ProAI are provided in Supplementary Movies 1.

## Statistical analysis

Dichotomous variables were reported as absolute numbers with percentages; continuous variables as mean (SD) or median (IQR). The Shapiro-Wilk test assessed distributional normality. Group comparisons utilized t-tests for normally distributed variables and Wilcoxon rank-sum tests for non-normal distributions. Low-risk/high-risk patient categorization employed t-tests, rank-sum tests, and chi-square tests as appropriate. Segmentation consistency was quantified using Dice similarity coefficient (DSC). Diagnostic performance evaluation encompassed area under the curve (AUC), sensitivity, specificity, accuracy, positive predictive value (PPV), and negative predictive value (NPV) at patient and lesion levels. The AUC was evaluated using continuous scores for both ProAI and PI-RADS. Sensitivity, specificity, accuracy, PPV, and NPV were computed using a probability threshold for ProAI optimized separately for each dataset by maximizing Youden's J statistic, and a fixed PI-RADS threshold of $\geq 3$. DeLong testing compared ProAI and reader performances, with statistical significance defined as $P < 0.05$. Analysis utilized SPSS (Statistics version 26.0, IBM Corp, Armonk, NY, USA) and R (version 4.6.0, R Foundation for Statistical Computing, USA). System specifications are detailed in Supplementary Note 10.

## Reporting summary

Further information on research design is available in the Nature Portfolio Reporting Summary linked to this article.

## Data availability

Source data underlying all figures and tables are provided with this paper (Source Data). The PI-CAI dataset used in this study is publicly available (https://pi-cai.grand-challenge.org). The TCIA cohort is publicly available via The Cancer Imaging Archive under the [Prostate-MRI-US-Biopsy] (DOI: 10.7937/TCIA.2020.A61IOC1A). De-identified derived data supporting the findings (for example, per-case labels, patient-level risk scores, and lesion-level masks exported from the trained model) have been deposited in [Figshare] (https://doi.org/10.6084/m9.figshare.28251827) and will be made publicly accessible upon publication. The raw clinical MRI data from participating hospitals contain protected health information and are available under restricted access to safeguard patient privacy and comply with institutional ethics approvals. Qualified academic investigators may request access for non-commercial research by contacting [gaoxu.changhai@foxmail.com; timchen91@aliyun.com; chengweishaoch@163.com; bianyun2012@foxmail.com] with (i) a brief proposal, (ii) evidence of local IRB/ethics approval or waiver, and (iii) a signed data-use agreement. Requests receive an initial response within 10 business days; if approved by the institutional data-access committee, data are shared via a secure platform. Unless legal or ethical obligations require otherwise, data will remain available for at least 5 years after publication. No genetic or other legally restricted data were generated. Any additional metadata needed to interpret, verify, and extend the findings are available in the paper, the Source Data file, or the cited repositories. Source data are provided with this paper.

## Code availability

Data are available through corresponding authors. Computer code is available as online inference codes on GitHub (https://github.com/lemon126/MRProstate). The ProAI model code has been made publicly available at DOI: 10.5281/zenodo.17301881.

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

## Acknowledgements

This work was supported in part by the National Science Foundation for Scientists of China (grant nos. 82171930 to Y.B., 82271972 to C.W.S., 82371955 to Y.B., and 62402501 to C.W.C.) and Clinical Research Plan of SHDC (grant nos. SHDC2022CRD028 to C.W.S.), Shanghai Municipal Health Commission Seed Program for Research and Translation of Medical New Technologies Project (grant nos. 2024ZZ1015 to Y.B.), and Plan for Promoting Scientific Research Paradigm Reform and Enhancing Disciplinary Advancement through Artificial Intelligence (grant nos. 2024RGZD001 to Y.B.) and Special Project for Clinical Research in the Health Industry of the Shanghai Municipal Health Commission (grant nos.202540148 to C.W.S). The funders had no role in the study design, data collection and analysis, decision to publish or preparation of the paper.

## Author contributions

Conceptualization: H.W., L.F., Y.B., C.W.S., J.W., X.G., J.P.; Data curation: H.W., L.F., Q.Y., H.C., Y.W., XG.Y., P.X., L.F., G.L., J.Y., Y.L., XD.Y.; Formal analysis: J.L., X.F.; Visualization: H.W., L.F., C.C., Y.B. J.L.; Writing-original draft: H., L.F., Y.B., C.W.S.; Technical support: J.L., Y.Y., X.C., C.C.; Funding acquisition: C.W.S., Y.B.

## Competing interests

The authors declare no competing interests.

## Informed consent

Informed consent was obtained for the publication of individually identifiable data reported in this study. For data where informed consent was not obtained, only aggregated information (e.g., averages) is presented to ensure no individual can be identified.
